# The Effect of Alkyl Substitution of Novel Imines on Their Supramolecular Organization, towards Photovoltaic Applications

**DOI:** 10.3390/polym13071043

**Published:** 2021-03-26

**Authors:** Paweł Nitschke, Bożena Jarząbek, Marharyta Vasylieva, Marcin Godzierz, Henryk Janeczek, Marta Musioł, Adrian Domiński

**Affiliations:** 1Centre of Polymer and Carbon Materials, Polish Academy of Sciences, 34 M. Curie-Skłodowska Str., 41-819 Zabrze, Poland; pnitschke@cmpw-pan.edu.pl (P.N.); mvasylieva@cmpw-pan.edu.pl (M.V.); mgodzierz@cmpw-pan.edu.pl (M.G.); hjaneczek@cmpw-pan.edu.pl (H.J.); mmusiol@cmpw-pan.edu.pl (M.M.); adominski@cmpw-pan.edu.pl (A.D.); 2Faculty of Chemistry, Silesian University of Technology, 9 Strzody Str., 44-100 Gliwice, Poland

**Keywords:** azomethines, supramolecular organization, organic thin films, polymer:fullerene blends, organic photovoltaics

## Abstract

Three novel conjugated polyazomethines have been obtained by polycondensation of diamines consisting of the diimine system, with either 2,5-bis(octyloxy)terephthalaldehyde or 9-(2-ethylhexyl)carbazole-3,6-dicarboxaldehyde. Partial replacement of bulky solubilizing substituents with the smaller side groups has allowed to investigate the effect of supramolecular organization. All obtained compounds have been subsequently identified using the NMR and FTIR spectroscopies and characterized by the thermogravimetric analysis, differential scanning calorimetry, cyclic voltammetry, UV–Vis spectroscopy, and X-ray diffraction. Investigated polymers have shown a good thermal stability and high glass transition temperatures. X-ray measurements have proven that partial replacement of octyloxy side chains with smaller methoxy groups induced a better planarization of macromolecule. Such modification has tuned the LUMO level of this molecule and caused a bathochromic shift of the lowest energy absorption band. On the contrary, imines consisting of N-ethylhexyl substituted carbazole units have not been so clearly affected by alkyl chain length modification. Photovoltaic activity of imines (acting as a donor) in bulk-heterojunction systems has been observed for almost all studied compounds, blended with the fullerene derivative (PCBM) in various weight ratios.

## 1. Introduction

From many years, a development of new, high-performance conjugated compounds and polymers has drawn much attention, due to their potential application in various optoelectronic systems, like organic photovoltaic cells (OPV), organic light emitting diodes (OLED), sensors, or organic field-effect transistors (OFET) [1,2,3,4]. Organic semiconductors used in these structures have revealed many advantages, compared to their inorganic counterparts, such as easy processing and low costs of production. The use of low-temperature, wet methods (like spin-coating, spray-coating or printing) of thin films deposition causes the significant reduction of production costs, compared to the chemical vapor deposition (CVD) or thermal vacuum evaporation (TVE) techniques. However, all these wet methods, require an appropriate solubility of used compounds. The possibility of obtaining large, elastics surfaces of organic materials, using wet methods, is also important from a practical point of view. Other advantage of organic compounds is the possibility of suitable modification of their chemical structure (by different substitutions or doping) to obtain desired properties and electronic structure, towards optoelectronic applications [5,6]. One of the interesting groups of organic semiconductors are azomethines and polyazomethines, known also as imines or Schiff-bases. Such materials are products of the condensation reaction between amines and aldehydes, where the formed imine bond (–C=N–) is proven to exhibit an isoelectronic character with a vinylene bond [7]. This condensation reaction is easy to conduct, in mild conditions, catalyzed by organic or inorganic acids, or even without using any [8,9], on the contrary to carbon-carbon or vinylene-coupled compounds, which usually require stringent reaction conditions and rather extensive purification processes [10]. Due to the relatively simple synthesis route and promising properties, conjugated azomethines and polyazomethines have been investigated for applications in optoelectronic systems, like photovoltaic solar cells [11,12,13], electroluminescent diodes [14,15], or electrochromic systems [16]. Highly aromatic polyazomethines are rather insoluble, so their thin films may be obtained via gas-phase condensation or by chemical vapor deposition (CVD) [17,18]. To ensure an appropriate solubility necessary for the thin films deposition by the spin coating method, a complexation of polyazomethine with the Lewis acid or di-m-cresol phosphate (DCP) [19], or much more often, a substitution with bulky side alkyl or alkoxy groups [20], may be used. It has been already proven that the introduction of alkoxy side groups modifies the electron structure of substituted compounds [21], while the length of alkyl chain only improves the solubility of materials [22]. However, such bulky side chains can affect the supramolecular organization as well [23], hindering the *π-*interactions, and subsequently decreasing a crystallinity and conductivity of compounds [24].

In this study, a partial replacement of bulky solubilizing substituents with smaller side groups has been investigated. Such an attitude is expected to ensure more favorable supramolecular organization, and as a result, a more planar arrangement of the macromolecules, which ought to exhibit more advantageous electrochemical and optical properties. To do so, three novel oligo- and polyazomethines have been obtained. Two of them have had an analogue chemical structure, which differed only in the length of part of the solubilizing side groups. The third imine has been substituted with both methoxy and octyloxy side groups and its properties have been compared with already reported polyazomethine with octyloxy side groups [11,22]. Thermal, electrochemical, and optical properties of these materials, together with the X-ray diffraction (XRD) patterns, have been thoroughly studied and discussed, in terms of the type of alkyl substituents. A final part of this paper presents the current-voltage (*J-V*) characteristics of bulk-heterojunction (BHJ) photovoltaic systems, utilizing these compounds, as donor materials, in a blend of polyazomethine (PAz) with fullerene acceptor (PC_61_BM) in various weight ratios (1:1, 1:2, 1:3). All these reported results provide new insights into the effect of supramolecular engineering on physicochemical properties of these novel compounds, towards their photovoltaic applications.

## 2. Materials and Methods

### 2.1. Materials

2,5-bis(octyloxy)terephthalaldehyde (98%), 9-(2-Ethylhexyl)carbazole-3,6-dicarboxa ldehyde (97%) and trifluoroacetic acid (TFA) (99%) have been purchased from Sigma-Aldrich and used as received. Diamines with diimine system, **DAAz1** and **DAAz2**, have been synthesized using a procedure described in [22]. The solvents such as toluene, methanol, chlorobenzene, and chloroform have been purchased from Avantor Performance Materials (Gliwice, Poland), and used as received. [6,6]-Phenyl-C61-butyric acid methyl ester (PC61BM) (>99% wt.) (M111) and PEDOT:PSS dispersion in water (M124) have been purchased in Ossila (Sheffield, UK) and used as received.

### 2.2. Characterization Methods

^1^H NMR spectra of synthesized polyazomethines have been recorded on Avance II Ultrashield Plus spectrometer, operating at 600 MHz using deuterated chloroform as a solvent and Tetramethylsilane (TMS) as an internal reference. The FTIR spectra have been recorded on JASCO FTIR 6700 Fourier transform infrared spectrometer, in a transmittance mode, in the range of 4000–400 cm^−1^ at a resolution of 2 cm^−1^ and for 64 accumulated scans. Size exclusion chromatography (SEC) has been performed in chloroform, at 35 °C with a flow rate of 1 mL/min, using a Spectra-Physic 8800 gel permeation chromatograph with a PL-gel 5 mm MIXED-C ultra-high efficiency column and Shodex SE 61 differential refractive index detector with polystyrene standards for calibration. DSC measurements have been taken with a DSC Q2000 apparatus (TA Instruments, Newcastle, DE, USA), in a range of −50–380 °C under the nitrogen atmosphere (flow rate was 50 mL/min), using aluminum sample pans. The instrument has been calibrated with a high-purity indium. In this study, the glass transition temperature (*T*_g_) has been taken as a midpoint of heat capacity change for amorphous samples obtained by quenching from melt in liquid nitrogen. Thermogravimetric analysis (TGA) has been performed with TGA/DSC1 Mettler-Toledo thermal analyses, in a range of 25 to 600 °C at a heating rate of 10°/min in a stream of nitrogen (60 mL/min). The obtained TGA data have been analyzed with the Mettler-Toledo Star System SW 9.30. The initial decomposition temperature has been taken as a temperature at the 5% weight loss (T_5%_). Electrochemical measurements have been performed on EDAQ E-corder 410 apparatus. The electrochemical cell has comprised of ITO (Indium Tin Oxide) quartz glass working electrode, an Ag|Ag^+^ electrode as a pseudo-reference electrode, and a platinum wire as an auxiliary electrode. A layer of polymer has been coated on the surface of the ITO plate. Measurements have been conducted at room temperature at a potential rate of 0.1 V/s. Electrochemical studies have been undertaken in 0.1 M solutions of Bu_4_NBF_4_, 99% (Sigma Aldrich, Saint Louis, MO, USA) in Acetonitrile (ACN). All electronic spectra of investigated polyazomethines have been measured with the two-beam JASCO V-570 UV-Vis-NIR spectrophotometer. The absorption spectra of solutions have been recorded in ranges of 240–800 nm (chloroform) or 200–2500 nm (thin films on quartz substrates). Concentration of all investigated solutions was 5 × 10^−4^ M. Thicknesses of spin-coated thin films have been measured using atomic force microscope, Topo-Metrix Explorer, working in the contact mode in the air, in the constant force regime. X-Ray diffraction studies have been performed using the D8 Advance diffractometer (Bruker, Karlsruhe, Germany) with Cu-Kα cathode (λ = 1.54 Å). Due to the critical angle for conjugated polymers using copper radiation being ~0.17° [25,26] and the layer thickness of sample (~100 nm), for 2D-GIWAXS setup, the 0.18° incidence angle has been applied, which is just above the critical angle for polymer layer and below the critical angle for glass support material. The scan rate has been 0.6°/min with scanning step 0.02° in range of 2.0° to 60° 2Θ (dwell time 2 s). Measurements have been performed in 7 variations, using different φ (Phi) angle, which corresponds to the sample rotation. As a φ = 0°, the longer edge has been set as parallel to the X-Ray beam direction. The resulting φ rotation (15, 30, 45, 60, 75, and 90°) has been programmed with a resolution of 0.1° φ. Obtained 2D patterns (with width of 3.1° 2θ) for different φ angle have been integrated to 1D patterns. Background subtraction, occurring from air scattering, has been performed using DIFFRAC.EVA program. All WAXD (XRD) measurements acquired at the different Phi angle have been accumulated to obtain the representative pattern, then obtained profiles have been smoothed, using a 5-point, quadratic polynomial, the Savitzky–Golay smoothing filter. For the structural analysis, the unit cell parameter *a* is related to the short macromolecule axis (correlated to the planarity and side chains) and *c* corresponds to the long axis of polymer (the length of macromolecule), while *b* is related to the π-stacking period [25].

### 2.3. Synthesis and Structural Characterization

Compounds investigated in this paper have been obtained by solution polycondensation of equimolar amounts of 2,5-bis(octyloxy)terephthalaldehyde and diamine **DAAz1**, consisting of diimine system, substituted with methoxy side chains, resulting in the formation of **PAz-BOO-OMe** (Figure 1). Polycondensation of equimolar amounts of 9-(2-Ethylhexyl)carbazole-3,6-dicarboxaldehyde with diamines consisting diimine system (**DAAz**), substituted with either methoxy (**DAAz1**) or octyloxy (**DAAz2**) side chains, has allowed to obtain **PAz-Carb-OMe** and **PAz-Carb-OOct,** respectively (Figure 1). The reagents have been dissolved in 3 mL of toluene (dried before use over magnesium sulfate) and provided with a magnetic dipole. The temperature has been increased up to 115 °C, while stirring the reaction mixture. After this, nitrogen has been passed through the reaction system to remove moisture and then 0.45 μL of trifluoroacetic acid has been introduced into the solution. The system has been sealed, leaving a flow of an inert gas (nitrogen) through it and the reaction has been carried out for two days, at 120 °C. After completion of the reaction, the product has been precipitated in methanol, dried in air, and then purified by Soxhlet extraction with methanol. The chemical structure of obtained compounds has been investigated using ^1^H-NMR, ^13^C-NMR, and FTIR spectroscopies.


**PAz-BOO-OMe**


**^1^H–NMR** (600 MHz, CDCl3, ppm) δ: 9.97 (s, C**H**O; end group), 9.91 (s, C**H**O; end group) 8.11 (1H, s, C**H**=N), 8.07–8.03 (1H, m, C**H**=N), 7.89–7.87 (1H, m, Ar–**H**), 7.82–7.77 (4H, m, Ar–**H**), 7.66–7.63 (1H, m, Ar–**H**), 7.47–7.36 (2H, m, Ar–**H**), 7.24 (1H, dd, J = 4.71, 3.95 Hz, Ar–**H**), 7.21–7.18 (2H, m Ar–**H**), 7.17–7.14 (1H, m, Ar–**H**), 7.10–7.08 (1H, m, Ar–**H**), 4.43 (2H, q, J = 7.15 Hz, –O–C**H_2_**–, ester), 4.26 (2H, q, J = 7.15 Hz, –O–C**H_2_**– ester), 2.98 (1H, s –O–C**H_3_**), 2.91 (1H, s –O–C**H_3_**), 1.50–1.43 (2H, m, –C**H_2_**– ether), 1.32 (3H, m,–C**H_2_** ether, –C**H_3_** ester, and ether). **^13^C–NMR** (150 MHz, CDCl3, ppm) δ: 165.57; 164.24; 160.27; 152.38; 147.02; 134.57; 130.86; 111.26; 69.13; 61.45; 60.39; 55.89; 31.78; 29.39; 25.95; 22.77; 14,29. **FTIR** (KBr, cm^−1^) υ: 3314 (N–H stretching), 2924, 2852 (C–H aliphatic stretching), 1728 (C=O stretching), 1688, 1677 (C=N, imine), 1578, 1409, 1366 (vibration of the thiophene rings), 1467 (vibration of the benzene rings), 1211 (C–O ether asymmetric stretching vibrations), adnd 1063 (C–O ether symmetric stretching vibrations).


**PAz-Carb-OMe**


**^1^H–NMR** (600 MHz, CDCl3, ppm) δ: 10.45 (s, C**H**O, end group), 10.16–10.09 (m, C**H**O; end group), 8.68 (2H, s, C**H**=N), 8.61 (1H, s, C**H**=N), 8.50 (1H, s, C**H**=N), 8.42–8.34 (1H, m, Ar–**H**), 8.23–8.19 (1H, m, Ar–**H**), 8.09 (1H, d, J = 8.28, Ar–**H**), 7.64 (1H, s, Ar–**H**), 7.54 (1H, d, J = 8.66 Hz, Ar–**H**), 7.50–7.41 (1H, m, Ar–**H**), 7.35 (1H, s, Ar–**H**), 4.50–4.45 (1H, m, –O–C**H_2_**–, ester), 4.44–4.37 (3H, m, –O–C**H_2_**– ester), 4.30–4.21 (6H, m, –O–C**H_2_**– ester), 3.95–3.86 (5H, m, –O–C**H_3_**), 1.51–1.20 (23H, m, –C**H_2_**– N–alkyl, –C**H_3_** ester), 0.96–0.91 (4H, m, –C**H_3_** N–alkyl), 0.87–0.82 (4H, m, –C**H_3_** N–alkyl). **^13^C–NMR** (150 MHz, CDCl3, ppm) δ: 165.57; 164.51; 160.00; 156.29; 153.11; 145.17; 129.80; 127.68; 124.24; 123.18; 110.20; 61.45; 60.39; 55.89; 48.20; 39.46; 30.98; 28.60; 24.36; 23.04; 14.29. **FTIR** (KBr, cm^−1^) υ: 3308 (N–H stretching), 2957, 2930, 2870 (C–H aliphatic, stretching), 1728 (C=O stretching), 1689, 1628 (C=N, imine), 1588, 1432, 1385 (vibration of the thiophene rings), 1535, 1475 (vibration of the benzene rings), 1253 (C–O ether asymmetric stretching vibrations), 1030 (C–O ether symmetric stretching vibrations).


**PAz-Carb-OOct**


**^1^H–NMR** (600 MHz, CDCl3, ppm) δ: 10.46 (s, C**H**O, end group), 10.15–10.11 (m, C**H**O; end group), 8.68 (2H, s, C**H**=N), 8.63–8.59 (1H, m, C**H**=N), 8.47–8.37 (1H, s, C**H**=N), 8.09 (1H, d, J = 8.66 Hz, Ar–**H**), 8.06–8.00 (1H, m, Ar–**H**), 7.65–7.56 (1H, m, Ar–**H**), 7.54 (1H, d, J = 8.28 Hz, Ar–**H**), 7.52–7.42 (1H, m, Ar–**H**), 7.32 (1H, s, Ar–**H**), 4.50–4.33 (4H, m, –O–C**H_2_**–, ester), 4.31–4.19 (5H, m, –O–C**H_2_**– ester), 4.10–3.99 (4H, m, –O–C**H_2_**– ether), 1.87–1.77 (2H, m, –CH_2_–C**H_2_**–, ether), 1.54–1.13 (37H, m, –C**H_2_**– N–alkyl, –C**H_3_** ester), 0.97–0.78 (11H, m, –C**H_3_** N–alkyl, ether). **^13^C–NMR** (150 MHz, CDCl3, ppm) δ: 165.57; 164.51; 160.00; 155.76; 153.11; 152.85; 145.17; 135.36; 129.80; 127.68; 124.24; 123.18; 110.20; 68.87; 61.45; 60.39; 48.20; 39.46; 31.78; 30.98; 29.66; 28.60; 26.21; 24.36; 22.77; 14.56; 14.03; 10.85. **FTIR** (KBr, cm-1) υ: 3425, 3310 (N–H stretching), 2955, 2926, 2855 (C–H aliphatic, stretching), 1731 (C=O stretching), 1677 (C=N, imine), 1589, 1424, 1385 (vibration of the thiophene rings), 1534, 1466 (vibration of the benzene rings), 1206 (C–O ether asymmetric stretching vibrations), 1027 (C–O ether symmetric stretching vibrations).

### 2.4. Organic Solar Cells Preparation

Devices with the bulk-heterojunction (BHJ) structure have been prepared on ITO-coated glass substrates (6 pixels, each with an area of 4.5 mm^2^, Ossila Ltd, Sheffield, UK). After cleaning the substrates with isopropanol in ultrasonic bath, a film of PEDOT:PSS has been deposited by spin coating. Solutions of the active layer have been prepared by dissolving blends of each individual polymer with the PCBM (1:1, 1:2 or 1:3 wt.) in chlorobenzene (previously dried over anhydrous magnesium sulfate). Such prepared solutions have been spin coated on the PEDOT:PSS layer and, subsequently, an aluminum counter electrode has been evaporated on the top of blend thin film. *J-V* curves of photovoltaic devices have been measured by the PV Test Solutions Solar Simulator, under the AM1.5 solar illumination and using the Keithley 2400 electrometer.

## 3. Results and Discussion

As shown in the literature [23], tailoring the solubilizing groups may greatly affect the molecular packing of materials through e.g., more favorable supramolecular organization. In this paper, such a modification has been achieved by partially replacing bulky solubilizing n-octyloxy groups with the shorter methoxy substituents. This modification of polyazomethine structure has been accomplished by the polymerization of diamines with diimine system, consisting of the methoxy substituted aromatic rings, with either 2,5-bis(octyloxy)terephthalaldehyde (**PAz-BOO-OMe**) or 9-(2-Ethylhexyl)carbazole-3,6-dicarboxaldehyde (**PAz-Carb-OMe** and **PAz-Carb-OOCt**). The presence of amine groups in the **DAAz** compounds has allowed to proceed with a copolymerization reaction with dialdehydes, as shown in the literature [27]. As reported in our previous research [11,22], the length of the alkyl chain in alkoxy substituent does not affect the absorption spectra and electrochemical properties of compounds, only their solubility, thermal properties [22], and photoluminescence intensity [11]. This is why the obtained polyazomethine **PAz-BOO-OMe**, consisting of both methoxy and octyloxy side chains, has been compared with an analogue polyazomethine, substituted solely with octyloxy side chains (**PAz-BOO-Oct**), previously reported in [22] (Figure 2). Apart from this, such an attempt on modification of supramolecular organization has been investigated on compounds consisting of branched N-ethylhexyl substituents, together with either methoxy (**PAz-Carb-OMe**) or octyloxy (**PAz-Carb-OOct**) side chains.

### 3.1. Structural and Solubility Studies

Chemical structure of obtained compounds has been confirmed using ^1^H– and ^13^C–NMR spectroscopies. Registered spectra have revealed signals, originating from aldehyde end groups in a range of 10.46–9.91 ppm; however, no signals of amine end groups have been observed. The presence of these signals might suggest the formation of rather oligomers than polymers, during the polycondensation reactions. Imine proton singlets have been observed in a range of 8.68–8.03 ppm, depending on the structure of imine. For **PAz-BOO-OMe**, only one group of these signals have been registered, between 8.11 and 8.03 ppm, while both **PAz-Carb** compounds spectra have revealed two groups of imine proton singlets, in ranges of 8.68–8.59 ppm and 8.50–8.37 ppm. This is due to various chemical environments of imine bonds in carbazole-consisting imines, where such bonds link together either thiophene and benzene pair or thiophene and carbazole pair. This is in contrary to **PAz-BOO-OMe**, where imine bonds link only thiophene and benzene rings, being only in one type of chemical environment. ^1^H-NMR spectrum of **PAz-BOO-OMe** has also revealed both singlets of methoxy substituents and multiplets originating from alkyl protons, present in octyloxy side chain. Similarly, spectra of both **PAz-Carb** imines have revealed signals of alkyl protons, present in the N-alkyl chains, together with either singlet of methoxy group (**PAz-Carb-OMe**) or multiplets from octyloxy side chains (**PAz-Carb-OOct**). Carbon spectra have revealed signals of imine groups in a range of 160.27–145.17 ppm. Methoxy carbons have given signal at 55.89 ppm, while methylene groups bonded with oxygen atoms in ether and ester side groups have given signals at higher chemical shifts, in a range of 69.13–60.39 ppm. Infrared spectra have revealed signals originating from amine end groups; however, they have been of very low intensity. Bands connected with imine stretching have been found in a range of 1689–1628 cm^−1^, while bands ascribed to stretching of carbonyl groups, present in aldehyde end groups and ester side chains, have been observed between 1731 and 1728 cm^−1^.

The solubility of compounds has been investigated in several organic solvents, of various dielectric constants and the results are presented in Table 1, together with the solubility test results of imine **PAz-BOO-OOct**, previously reported in [22].

None of investigated compounds has been soluble in a non-polar n-hexane. Imines consisting only bulky substituents (**PAz-BOO-OOct**, **PAz-Carb-OOct**) have been completely or mostly soluble in the remaining solvents, both in polar NMP and also in chloroform or THF of lower dielectric constants. Partial or complete replacement of bulky side chains with methoxy groups (**PAz-BOO-OMe** and **PAz-Carb-OMe,** respectively) has drastically decreased the solubility of compounds in NMP and THF and hindered their affinity to chloroform. Observed solubility in this solvent has, nevertheless, been sufficient for the purpose of further investigations.

The molar masses and dispersity values have been calculated according to the conventional calibration with polystyrene standards. Obtained values are presented in Table 2 together with molar masses of **PAz-BOO-OOct**, reported previously in [22]. The presented values have proved that the synthesis of imine **PAz-BOO-OMe** has resulted in a formation of oligomer, while the remaining polycondensations have allowed to obtain compounds with a higher degree of polymerisation.

Imines consisting of methoxy substituents have reached lower molar masses than their counterparts consisting solely of bulky side groups. Similarly, compounds with branched N-ethylhexyl chains have revealed higher molar masses than analogue imines with octyloxy side chains. This is probably due to the better solubility of forming macromolecules, consisting of such substituents, allowing the formation of longer polymer chains with higher molar masses. The compound **PAz-Carb-OOct**, consisting of both types of bulky substituents, has shown the highest molar mass of all investigated imines.

### 3.2. Thermal Properties

Glass transition temperatures from differential scanning calorimetry (DSC) measurements and weight loss data, obtained from thermal gravimetric analysis (TGA) of synthesised polymers, together with analogue values, previously reported in [22] for **PAz-BOO-OOct**, are gathered in Table 3. The DSC curves obtained during the first heating scans of investigated polyazomethines have not revealed endotherms related to melting, and during the second heating stage, after a rapid cooling, all have shown a characteristic deflection, connected to the glass-transition (Appendix A).

The analysis of registered parameters has revealed that utilization of diamines with diimine system has allowed to obtain compounds with much higher glass transition temperatures, in a range of 264.0–318.0 °C, compared to the imine, synthesized from the simpler monomers (**PAz-BOO-OOct**), which have shown *T_g_* = 30.8 °C. Such a difference may be due to the large amount of long n-alkyl chains, together with a moderate oligomer chain length. All investigated compounds have shown a one-step thermal degradation process, with similar initial decomposition temperatures (Appendix A). Unfortunately, due to the low amount of **PAz-BOO-OMe**, TGA measurements have not been completed for this compound. Designated T_5%_ values have been observed to be lower for imines with branched ethylhexyl side chains (**PAz-Carb**) and have been the lowest for polyazomethine consisting solely of bulky substituents (**PAz-Carb-OOct**). The temperatures of 10% mass loss have been almost identical for all investigated compounds and have probably been connected to the thermal degradation of ester side chains, present in all of those structures.

### 3.3. Electrochemical Measurements

Electrochemical measurements have been conducted using the cyclic voltammetry (CV) technique, performed in the range of all noticeable oxidation and reduction peaks, within the range of electrolyte stability window. Obtained voltammograms have revealed that both oxidation and reduction processes, of all investigated polyazomethines, have been electrochemically irreversible (Figure 3a). The onsets of electrochemical processes have been designated, which has allowed to calculate the energies of HOMO (the Highest Occupied Molecular Orbital) and LUMO (the Lowest Unoccupied Molecular Orbital) levels, together with the energy gaps ($E_{g}^{CV}$). All of these parameters have been gathered in Table 4, and presented in Figure 3b, together with the values previously reported for **PAz-BOO-Oct** [11].

Analysis of presented values of **PAz-BOO** imines has revealed an increase of oxidation onset potential and a simultaneous decrease of the *E*_red_^onset^, upon a partial replacement of octyloxy side chains (**PAz-BOO-OOct**) with shorter methoxy substituents (**PAz-BOO-OMe**). The reduction potential, nevertheless, has been decreased more significantly than the oxidation potential has been increased, which has resulted in more lowered LUMO orbital and, subsequently, in a decrease of the band gap width by 0.25 eV of methoxy-substituted imine, compared to the analogue compound, consisting solely of bulky side chains. Since **PAz-BOO-OMe** has had lower molar mass, which has also an impact on electrochemical properties [29], this is most probably due to adopting a more planar geometry by the oligoimine **PAz-BOO-OMe**, due to the lower amount of bulky alkyl chains. Such planarity adjustment has tuned the LUMO level of compound, with the slight effect on HOMO level [30]. Similar manipulation of substituents′ length has almost not affected either oxidation or reduction potentials of carbazole-consisting compounds (**PAz-Carb**). They have undergone these processes at nearly identical potentials and have subsequently revealed very similar energies of HOMO and LUMO orbitals and similar energy gap widths. This might suggest that disruption of macromolecule planarity, caused by branched N-ethyl-hexyl chains, cannot be influenced by the manipulation with alkoxy substituent lengths.

### 3.4. Optical Properties

UV-Vis absorption spectra have been measured for imines solutions in chloroform and for their thin films, deposited on quartz substrates, using the spin-coating technique. Electronic spectra of investigated oligo– and polyazomethines solutions (Figure 4a) have shown absorption bands in a range of 263.5–684.5 nm. The bands localized at the lowest energies have been attributed to the *π → π** electronic transitions. Absorption bands localised at higher energies are due to the electron transitions between the *σ → π** or *π → σ** levels. Position of the low energy absorption bands have been influenced by the length of alkoxy substituent and by the chemical structure of imine, while bands registered at shorter wavelengths have revealed similar positions for all compounds, although their intensity have been much higher for **PAz-Carb** imines.

The comparison of oligoazomethine **PAz-BOO-OMe** spectrum with that recorded previously for polyazomethine **PAz-BOO-OOct** [22] has revealed a distinct bathochromic shift of the absorption band connected with *π → π** transitions (from 568.0 to 684.5 nm). Due to a lower molar mass of this compound, such a shift could have not resulted from larger *π*-conjugation area on longer polymer chains [31], which is consistent with the electrochemical data, and most probably has been due to adopting the more planar geometry by this molecule. Similar manipulation with substituents′ length in **PAz-Carb** imines has resulted in a much smaller bathochromic shift of the band (from 446.5 to 513.0 nm), after the complete replacement of octyloxy substituents (**PAz-Carb-OOct**) with methoxy side groups (**PAz-Carb-OMe**). This might also suggest an increase of the molecule planarity, upon modification of side chains length, although in a much smaller extent.

Absorbance spectra of thin films of investigated compounds (Figure 4b) have revealed additional bands, localized at high energies (from 212.5–220.0 nm), which are connected to the *σ → σ** electron transitions and could have not been observed in solution, due to the solvent measurement window (the spectral range of solvent permeability). Almost all imines have shown a bathochromic shift of the absorption band connected with the *π → π** electron transitions, after deposition of thin film, compared to the solution (Table 5). This suggests the formation of *J*-type aggregates during spin-coating [32]. Unlike others, absorption band of **PAz-BOO-OMe** thin film has shifted hypsochromically, which is probably due to formation of *H*-type aggregates in this individual imine thin film [32]. The absorption bands, nevertheless, have been still observed at lower energies for imines consisting of methoxy side groups, than for their octyloxy-substituted analogues.

Moreover, all absorption bands connected with the *π* → *π** electron transitions have shown a vibronic structure (Figure 5). Low energy bands of the methoxy-substituted compounds (**PAz-BOO-OMe** and **PAz-Carb-OMe**) have shown a clear vibronic structure, while individual vibronic peaks of remaining imines have been found using the second derivative method (i.e., minimum of the second derivative of absorption corresponds to the absorption maximum). Afterwards, the vibronic progression of imine thin films bands have been deconvoluted, with the modified Fourier self-deconvolution and Finite Response Operator (FIRO) methods [33]. Observed vibronic peaks have been assigned according to the Franck-Condon principle, assuming the stationary nuclear framework [34]. Energy differences between individual vibronic peaks have been in the range of 0.16–0.25 eV, which clearly suggests presence of the electron–phonon interaction, which are connected to the benzene ring stretching mode [35].

Spectra of **PAz-BOO** thin films (Figure 5a) have revealed the most pronounced peaks connected to transitions between *0-0* and *0-1* levels, while peaks connected to transitions of higher energies have been of much smaller intensity. In contrary, deconvoluted spectra of **PAz-Carb** imines (Figure 5b) have shown a gradual increase of intensity as the vibronic peak energy increased. Assuming that the main intramolecular vibration (*E_p_*), coupled to the electronic transition, is the C=C symmetric stretch at 0.18 eV [36], a ratio of intensities of *0-0* and *0-1* vibronic peaks has allowed to calculate values of exciton bandwidths (*W*) for each of individual thin film, according to the Equation (1) [37].
(1)$$\frac{A_{0-0}}{A_{0-1}}=\;\frac{1-0.24W/E_{p}}{1+0.073W/E_{p}}$$

All values of calculated exciton bandwidths (*W*) are gathered in Table 5. This parameter is connected to the effective conjugation length, where an increase of the conjugation will lead to a decrease of exciton bandwidth [38]. Such a correlation, nonetheless, is only true in systems with similar interchain order, thus calculated values have been compared only within systems with analogue chemical structure. For both groups that are **PAz-BOO** and **PAz-Carb** imines, the value of *W* has been much smaller for compounds consisting of methoxy side groups (**PAz-OMe**) than for their octyloxy-substituted counterparts (**PAz-OOct**). This clearly indicates that shortening of the alkyl chain length has tuned the macromolecule geometry, allowing for. adopting the more planar configuration, with an enhanced effective *π*-conjugated length 

The edges of absorption bands, connected to the *π → π** electron transitions, of investigated imines thin films have allowed to determine absorption edge parameters, which is the Urbach energy (*E_U_*) and the energy gap width (*E_g_*). Values of the Urbach energy have been calculated based on the slope of exponential edges, which follow the Urbach relation (2) [39], as it is depicted in Figure 6a.
(2)$$\alpha \propto exp\;(\frac{E}{E_{U}})$$

Energy gaps have been obtained, using a linear approximation to energy axis of the *(αE*)^1/2^ dependence, according to the Tauc relation (3) [40]:(3)$$\alpha \propto {\left(E-E_{g}\right)}^{2}$$

(true for the energy range *E > E_g_*), typical for amorphous semiconductors, often used also for polymers thin films [17,18,21,22]. The way of determination the energy gaps for investigated films is shown in Figure 6b. All obtained absorption edge parameters have been gathered in Table 5.

Generally, carbazole consisting imines have shown higher values of the Urbach energy, indicating a higher amount of structural disorder defects, which have introduced localized energy states within the energy gap. Especially the absorption edge of **PAz-Carb-OMe** thin film has shown high value of *E_U_* = 349 meV, suggesting the large number of localized states within the energy gap. All energy gap values obtained using the Tauc model, have been lower than these designated using cyclic voltammetry. Compounds **PAz-BOO** have shown rather close values of energy gaps calculated by these two methods, and a partial replacement of octyloxy side chains with methoxy groups (**PAz-BOO-OMe**) has caused a decrease of *E_g_*, compared to imine solely substituted with bulky alkyl chains (**PAz-BOO-OOct**), from 1.69 to 1.59 eV. Energy gap width of **PAz-Carb-OOct** has been also very similar (1.77 eV) to that, obtained by electrochemical measurements, although the replacement of octyloxy chains with methoxy groups (**PAz-Carb-OMe**) has caused a significant decrease of *E_g_*, to a value much lower (1.42 eV) than the electrochemically determined. Such a low value has been most probably caused by the high amount of localized states within the energy gap, which allows energy transitions, of lower energy than the energy gap width.

### 3.5. Morphology of Thin Films

Morphology of investigated imines thin films has been observed using the Grazing-Incidence Wide-Angle X-ray Scattering (GIWAXS) measurements. Obtained diffractograms (Figure 7) have revealed that most of the investigated imines have shown rather low-ordered structure, showing only one broad signal, localized at similar positions, between 7.58°–8.22° 2θ, with a small broadening in a range of 4.74°–5.02° 2θ, which indicates a presence of weak signal. This peak has become a clearer signal for **PAz-BOO-Oct**. Only imine **PAz-BOO-OMe** has shown a much more ordered structure with multiple peaks.

The structure order, visible in the form of peaks in WAXD patterns, is related to the macromolecules planarity and intermolecular interactions, but is not always related to π-stacking peak. In the case of thin-film examination, a presence of *π*-stacking peak indicates a highly ordered structure. In ordered structures, with few peaks visible, more planar molecules might be characterized by a (100) peak position—higher *d*-spacing values show higher macromolecule planarity [25,26]. After the assignment of Miller indexes, it has been noticed that both carbazole-consisting imines (***PAz-Carb***) have shown only a peak corresponding to the *a* parameter, connected to the short polymer axis, and thus with the planarity of macromolecules. Almost identical positions of these peaks (7.6° and 7.7° 2θ for **PAz-Carb-OOct** and **PAz-Carb-OMe,** respectively) has indicated lack of any changes in the molecule geometry, regardless of alkyl chain variation. This is in agreement with the electrochemical data, registered for these compounds. Polyazomethine substituted solely with octyloxy groups (**PAz-BOO-Oct**) has revealed that the peak, corresponding to planarity of the molecule, shifted towards higher angles (8.2° 2θ), suggesting a narrower a axis dimensions, which may be connected to the presence of linear, n-octyloxy side chains. Apart from that, a peak ascribed to the c parameter, connected to the length of molecules, at 4.8° 2θ, has become more developed. The partial replacement of bulky octyloxy groups with methoxy substituents (**PAz-BOO-OMe**) has resulted in a significant increase of solid order. A peak, ascribed to the c parameter, has been visible at lower angles (2.8° 2θ), suggesting larger dimensions in this axis. Moreover, the diffractogram of this imine thin film has shown peaks ascribed to multiples of this parameter (5.3° and 8.8° 2θ). The signal corresponding to the planarity of macromolecule, assigned to the a parameter, has been observed at lower angles for **PAz-BOO-OMe** (3.6° 2θ), compared to **PAz-BOO-Oct**, (8.2° 2θ). This clearly suggests the larger dimension in this axis, which suggests the enhanced planarity of this imine in respect to its counterpart substituted solely with octyloxy side chains. According to [25,26], spin-coated thin films exhibits lower order than solid samples, especially for high molar masses of macromolecules.

### 3.6. Preliminary Photovoltaic Activity Tests

All of the investigated compounds (PAz) have been utilized in photovoltaic bulk-heterojunction (BHJ) systems, acting as donor, together with the fullerene derivative (PC_61_BM), acting as acceptor. The conventional architecture ITO/PEDOT:PSS/PAz:PC_61_BM/Al has been chosen, where various weight ratios of donor and acceptor have been studied (1:1, 1:2 and 1:3 wt.). Registered *J-V* characteristics (Figure 8) have allowed to designate parameters of prepared photovoltaic cells (Table 6). 

Almost all of the studied compounds have shown a photovoltaic effect, while acting as a donor, with [6,6]-phenyl-C61-butyric acid methyl ester (PCBM) as an acceptor, except for **PAz-Carb-OMe**, which has shown no activity in all systems. Lack of any photo-response of this compound may be caused by a large amount of structural disorder effects, which have introduced a localized energy states within energy gap and are trapping generated charge carriers. Power conversion efficiencies of studied systems have been in the range of 0.02–0.17%. Such values are similar to others, reported for BHJ systems with phenylene or thiophene-phenylene imines, described in the literature [41,42]. The highest power conversion efficiency has been observed for systems consisting of **PAz-BOO-Oct** imine. Partial replacement of octyloxy side chains with methoxy groups (**PAz-BOO-OMe**), despite providing more favorable optical and electrochemical properties, has decreased efficiency of photovoltaic cell, where modified imine has been utilized. The main parameter, which has been responsible for such a decrease, has been the short-circuit current density (*J_SC_*). A decrease of this parameter has probably been most connected with the lower molar mass of the **PAz-BOO-OMe.** For shorter macromolecules, the ratio of intermolecular charge transport in relation to the intramolecular part is higher, which causes a decrease of the overall conductivity in the material [43]. The increase of the donor:acceptor ratio to 1:2 has enhanced the efficiency of photovoltaic systems consisting **PAz-BOO-OOct** and **PAz-Carb-OOct**, while it has completely ceased any activity of **PAz-BOO-OMe**. Further increase of the acceptor quantity has ceased activity of the remaining imines.

## 4. Conclusions

In this paper, three novel oligo- and polyazomethines have been obtained and the influence of alkyl side chains lengths on their supramolecular organization has been observed. Such a modification has been accomplished by the condensation of diamines with diimine systems that, to the best of our knowledge, is presented for the first time in literature.

The realization of such an approach has been confirmed using ^1^H–, ^13^C–NMR, and FTIR spectroscopies. All of new compounds have shown good thermal stability, and high glass transition temperatures. They have been electrochemically active, and revealed narrow energy gaps, being in the range of 1.64–1.87 eV. Electronic spectra of all new imines solutions and thin films have revealed the broad absorption range. During the spin-coating process of most imines, the *J* type aggregates have been formed, except for the compound with both octyloxy and methoxy side groups. The partial replacement of the bulky octyloxy side chains with the shorter methoxy groups has induced the adoption of a more planar geometry by a macromolecule, tuning the LUMO orbital and subsequently decreasing the energy gap. This has provided more favorable optical properties, in terms of application in photovoltaic systems, shifting the absorption band connected with the *π → π** electron transitions. Variation of the alkoxy side chains length, nevertheless, has not influenced the electrochemical properties and only slightly affected electronic spectra, when branched N-ethylhexyl chains have been present in the polymer structure.

Almost all compounds have shown the activity in photovoltaic devices, acting as a donor in the blend with fullerene. It has been demonstrated that the chemical structure of investigated imines (both the main group, as side chains) determined the photovoltaic properties, but also the ratio polymer:fullerene in the BHJ active layers is very important in these organic solar cells. Generally, carbazole consisting imines have exhibited a worse photovoltaic properties than their thiophene-phenylene counterparts, while the presence of octyloxy side chains enhance these properties for both groups of imine compounds.

This paper has shown that through the variation of substituents length, it is possible to change the supramolecular structure, which influences the electrochemical and optical properties of materials. Such an approach provides useful information, which may be used during the designing of novel compounds with properties, desired for the application in optoelectronic systems.

## Figures and Tables

**Figure 1 polymers-13-01043-f001:**
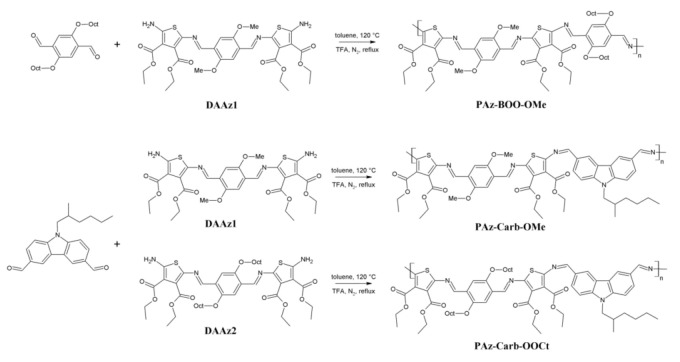
Synthesis procedure and chemical structures of investigated polyazomethines.

**Figure 2 polymers-13-01043-f002:**
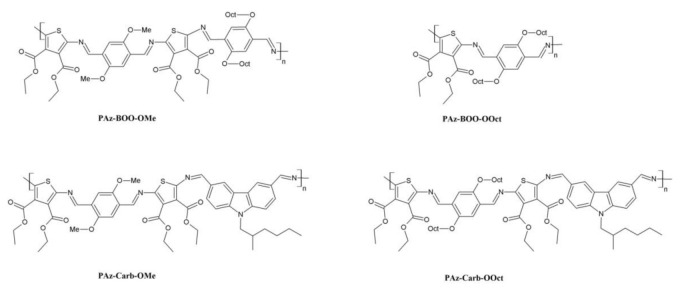
Chemical structures of synthesized compounds, together with polyazomethine obtained during previous studies (**PAz-BOO-Oct**) [22].

**Figure 3 polymers-13-01043-f003:**
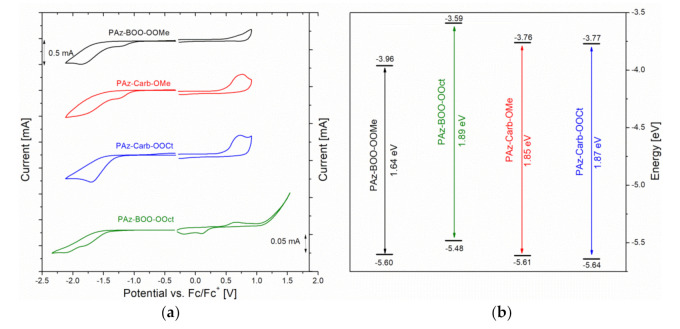
(**a**) Voltammograms, recorded during oxidation and reduction processes (v = 0.1 V/s; 0.1 M Bu_4_NPF_6_/ACN) of studied oligo- and polyazomethine thin films, deposited on the glass with ITO together with voltammogram obtain for **PAz-BOO-Oct**, presented previously in [11]; (**b**) the energy levels of HOMO and LUMO orbitals, designated based on the results of cyclic voltammetry measurements.

**Figure 4 polymers-13-01043-f004:**
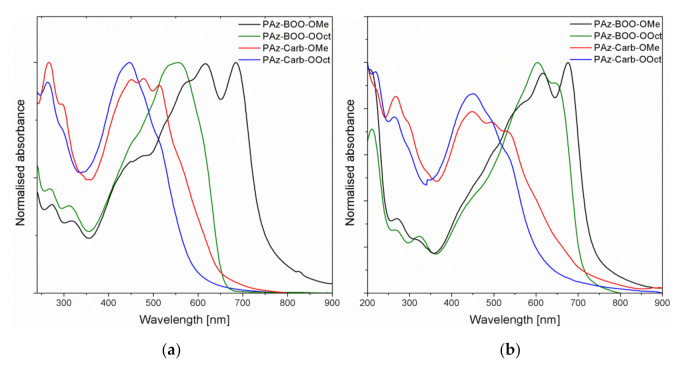
(**a**) Normalized absorbance of investigated oligo- and polyazomethines solutions in chloroform and (**b**) their thin films, deposited on quartz substrates.

**Figure 5 polymers-13-01043-f005:**
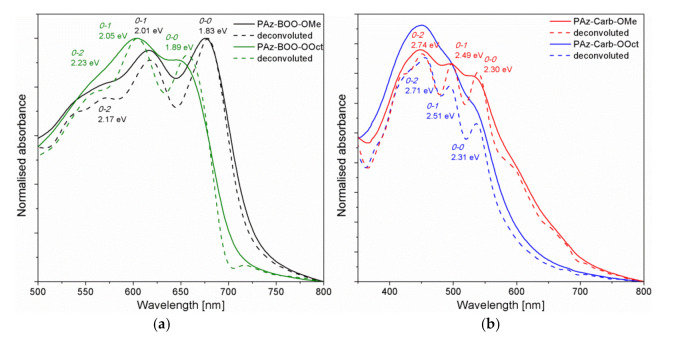
Normalized (solid line) and deconvoluted (dashed line) spectra of **PAz-BOO** (**a**) and **Paz-Carb** (**b**) imine thin films.

**Figure 6 polymers-13-01043-f006:**
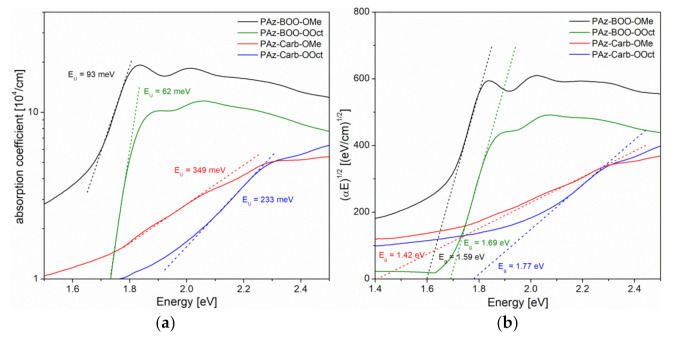
Absorption edge parameters of investigated imines: (**a**) Urbach energies and (**b**) energy gaps.

**Figure 7 polymers-13-01043-f007:**
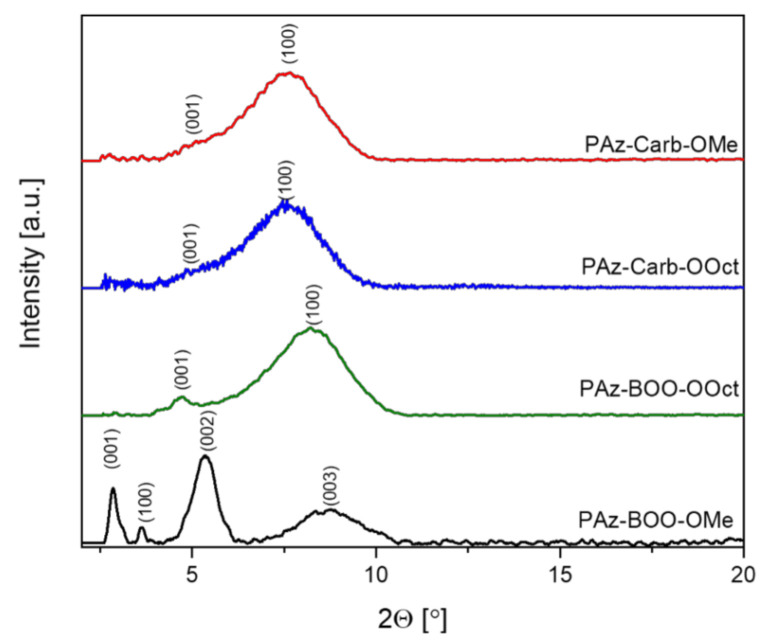
XRD patterns of imine thin films with possible assignment of Miller indexes.

**Figure 8 polymers-13-01043-f008:**
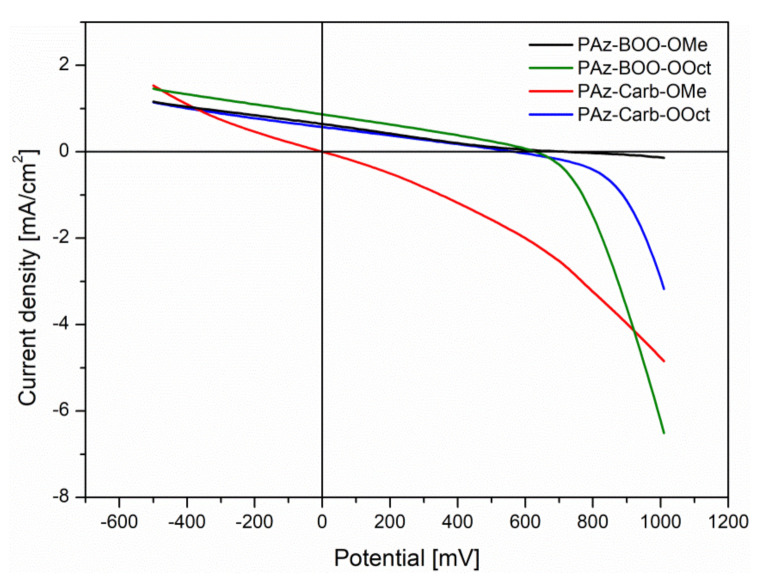
Current density–voltage (*J-V*) characteristics of bulk-heterojunction (BHJ) systems, consisting investigated oligo- and polyazomethines. Donor:acceptor ratios 1:2 for all, except for **PAz-BOO-OMe**, where the ratio D:A is 1:1.

**Table 1 polymers-13-01043-t001:** Solubility of investigated compounds in solvents of various dielectric constants (ε).

Compound	NMP (ε = 33.00)	THF (ε = 7.58)	Chloroform (ε = 4.80)	n-Hexane (ε = 1.88)
**PAz-BOO-OMe**	-/-	-/-	+/-	-/-
**PAz-BOO-OOct**	+/- *	+/+ *	+/+ *	-/- *
**PAz-Carb-OMe**	-/-	-/-	+/-	-/-
**PAz-Carb-OOct**	+/+	+/+	+/+	-/-

1 mg/mL (+/+) soluble at room temperature, (+/-) soluble at boiling point, (-/-) partially soluble or insoluble at boiling point. ***** values from [22].

**Table 2 polymers-13-01043-t002:** Molar masses and molar mass dispersity of investigated polyazomethines.

Compound	M_n_ [g/mol]	M_w_ [g/mol]	*Ð*
**PAz-BOO-OMe**	1350	3250	2.4
**PAz-BOO-OOct**	2690 *	7520 *	2.8 *
**PAz-Carb-OMe**	4700	5300	1.2
**PAz-Carb-OOct**	9600	16,350	1.7

M_n_—number average molar mass, M_w_—mass average molar mass, *Ð*—molar mass dispersity, ***** values from [22].

**Table 3 polymers-13-01043-t003:** Thermal properties of synthesized polymers.

Compound	*T*_g_ [°C]	*T*_5%_ [°C]	*T*_10%_ [°C]	*T_max_* [°C]
**PAz-BOO-OMe**	318.0	n/d	n/d	n/d
**PAz-BOO-OOct**	30.8 *	350.0 *	364.8 *	380.0 *
**PAz-Carb-OMe**	264.0	341.5	368.8	391.5
**PAz-Carb-OOct**	291.0	331.3	365.5	393.7

*T_g_*—glass transition temperature, *T*_5%_, *T*_10%_—temperatures of 5% and 10% weight loss, *T_max_*—temperature of the maximum mass loss rate, n/d—not designated, ***** values from [22].

**Table 4 polymers-13-01043-t004:** Electrochemical properties of investigated compounds.

Compound	*E*_ox_^onset^ [V]	*E*_red_^onset^ [V]	*E*_HOMO_ [eV]	*E*_LUMO_ [eV]	*E*_g_^CV^ [eV]
**PAz-BOO-OMe**	0.50	−1.14	−5.60	−3.96	1.64
**PAz-BOO-OOct**	0.38 *	−1.51 *	−5.48 *	−3.59 *	1.89 *
**PAz-Carb-OMe**	0.51	−1.34	−5.61	−3.76	1.85
**PAz-Carb-OOct**	0.54	−1.33	−5.64	−3.77	1.87

*E*_ox_^onset^—the onset of oxidation potential, *E*_red_^onset^—the onset of reduction potential, *E*_HOMO_ = −e^−^(5.1 + *E*_ox_^onset^), ***E*_LUMO_** = −e^−^(5.1 + *E*_red_^onset^) according to [28], ***E*_g_^CV^**—electrochemical energy gap = *E*_LUMO_-*E*_HOMO__,_ * values from [11].

**Table 5 polymers-13-01043-t005:** Positions of absorption bands connected with *π → π** transitions of imine solutions and thin films, together with exciton band widths and absorption edge parameters of thin films.

Compound	*λ_max_* [nm]/(*E*_max_ [eV])		*W* [meV]	*E*g [eV]	*E*_U_ [meV]
	Solution	Thin Film			
**PAz-BOO-OMe**	684.5/(1.81)	676.0/(1.83)	4	1.59	93
**PAz-BOO-OOct**	556.0/(2.23) *	604.0/(2.05) *	46	1.69 *	62 *
**PAz-Carb-OMe**	513.0/(2.42)	535.0/(2.32)	30	1.42	349
**PAz-Carb-OOct**	446.5/(2.78)	450.0/(2.76)	85	1.77	233

*λ_max_*—position of lowest-energy distinct absorption band*, W*–exciton bandwidth, *Eg*—energy gap designated according to Tauc model, *E_U_*—Urbach energy, ***** values from [22].

**Table 6 polymers-13-01043-t006:** Photovoltaic parameters of the BHJ photovoltaic systems, consisting investigated imines.

System	*V_OC_* [mV]	*J_SC_* [mA/cm2]	*FF*	*η* [%]
**PAz-BOO-OMe**:PCBM (1:1)	728.2	0.65	0.20	0.09
**PAz-BOO-OMe**:PCBM (1:2)	less than 0.01%			
**PAz-BOO-OMe**:PCBM (1:3)				
**PAz-BOO-Oct**:PCBM (1:1)	834.8	0.72	0.16	0.16
**PAz-BOO-Oct**:PCBM (1:2)	630.8	0.87	0.29	0.17
**PAz-BOO-Oct**:PCBM (1:3)	less than 0.01%			
**PAz-Carb-OMe**:PCBM	less than 0.01% for each ratio			
**PAz-Carb-OOct**:PCBM (1:1)	210.6	0.22	0.47	0.02
**PAz-Carb-OOct**:PCBM (1:2)	615.1	0.58	0.26	0.09
**PAz-Carb-OOct**:PCBM (1:3)	less than 0.01%			

*V_OC_*-open circuit voltage, *J_SC_*—short circuit current density, *FF*—fill factor, *η*—power conversion efficiency.

## Data Availability

The data presented in this study are available on request from the corresponding author.

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
