# Peer review of "The Effect of Alkyl Substitution of Novel Imines on Their Supramolecular Organization, towards Photovoltaic Applications"

_polymers, 2021, doi:10.3390/polym13071043_

Round 1

Reviewer 1 Report

This work synthesizes three new polymers through polycondensation of diamines with the aldehyde. The morphological characterization was performed, and optoelectronic properties were characterized. However, the morphological study has many confusing discussions with limited raw data proving the analysis. The PCE values here are ~40 times lower when compared with the traditional P3HT/PCBM blend. The value of this work to the field is questionable.

  1. In Section 3.2, the raw data for DSC and TGA is not reported. Please prepare this information in the supplementary information for further review. Weirdly, Paz-Boo-OOct shows a much lower Tg when compared with other polymers.
  2. Table 5 lists the position of the lowest-energy absorption band for all four polymers. Please check the values for PAz-BOO-OMe and PAz-BOO-OOct under the thin-film state. It seems the position of 0-0 and 0-1 transition was mixed up. If that is the case, the other values need to be double-checked again.
  3. The assignment of Miller indices needs more explanation. Why is the peak at around 7.5 degrees named as (100) for the first three polymers, while it was named (003) for Paz-BOO-OMe? Moreover, how does the author attribute (001) and (100) to the Paz-BOO-OMe polymer? It is suggested to provide a more detailed explanation or draw a diagram of the packing phenomenon. The origin of these peaks is not clear from the description.
  4. Line 482: The author claimed: both carbazole-consisting imines (PAz-Carb) have shown only a peak corresponding to a parameter, connected with the short polymer axis, and thus with the planarity of macromolecules. Is this “short polymer axis” correlated with side-chains or the width of the chain? How does the short polymer axis correlate with the planarity of the macromolecules? Any references for it?
  5. Line 488: The author claimed: Polyazomethine substituted solely with octyloxy groups (PAz-BOO-Oct) has revealed the peak, corresponding to  planarity of the molecule, shifted towards higher angles (8.2° 2θ), suggesting wider an axis dimensions, which may be connected to the presence of n-octyloxy side chains. What is “an axis”? If the author means “a axis”, then it is confusing. An increase in 2theta value corresponds to a shorter correlation length. What does the author mean by “a wider a axis dimension”?
  6. It is not clear how the authors define the observation of the “planar” structure from WAXD. In conjugated polymer systems, one way to determine it is by observing the pi-pi stacking peak, which is typically at ~20 – 25 degrees and is missing from the data presented here.
  7. The optoelectronic properties were measured in the blend system with PCBM, but the morphology for the blend system was not investigated. There is a lack of systematic study on the morphology-performance relationship.

Author Response

Reviewer #1:

This work synthesizes three new polymers through polycondensation of diamines with the aldehyde. The morphological characterization was performed, and optoelectronic properties were characterized. However, the morphological study has many confusing discussions with limited raw data proving the analysis. The PCE values here are ~40 times lower when compared with the traditional P3HT/PCBM blend. The value of this work to the field is questionable.

REPLY: It is true, that the efficiency of investigated systems is significantly lower than these of traditional polythiophene/fullerene systems. However, reported values are typical for these systems, consisting of polyazomethines. To support this, a following sentence:

Power conversion efficiencies of studied systems have been in a range 0,02 – 0,17%. Such values are similar to others, reported for BHJ systems with phenylene or thiophene-phenylene imines, described in a literature [41,42].

 has been added into a chapter 3.6. Preliminary photovoltaic activity tests, together with appropriate, new references [41,42].

  1. In Section 3.2, the raw data for DSC and TGA is not reported. Please prepare this information in the supplementary information for further review. Weirdly, Paz-Boo-OOct shows a much lower Tg when compared with other polymers.

REPLY: DSC, TGA and DTG curves of studied compounds have been gathered and presented in Supplementary Information (Figures S1 and S2) and appropriate references in the main text have been introduced. Indeed, there is a surprising difference between Tg of novel compounds, and that reported previously. However, all values are, in a range acceptable for polyazomethines and additionally the following sentence:

Such difference may be due to the large amount of long n-alkyl chains, together with a moderate oligomer chain length.

has been added to the part 3.3 Thermal properties.

  1. 2. Table 5 lists the position of the lowest-energy absorption band for all four polymers. Please check the values for PAz-BOO-OMe and PAz-BOO-OOct under the thin-film state. It seems the position of 0-0 and 0-1 transition was mixed up. If that is the case, the other values need to be double-checked again.

REPLY: Thank you for this comment. The value of 0-0 vibronic band energy of PAz-Oct-OMe presented in Table 5 has not matched the value presented in Figure 5. The wavelength values and remaining energies have been double-checked. In Table 5 are presented only positions of a distinct, lowest energy absorption bands. This is not always in agreement with position of the 0-0 vibronic peak, since this peak occurs in many cases as a deflection, not as a distinct band. Appropriate modification of Table 5 footnote has been performed.

  1. The assignment of Miller indices needs more explanation. Why is the peak at around 7.5 degrees named as (100) for the first three polymers, while it was named (003) for Paz-BOO-OMe? Moreover, how does the author attribute (001) and (100) to the Paz-BOO-OMe polymer? It is suggested to provide a more detailed explanation or draw a diagram of the packing phenomenon. The origin of these peaks is not clear from the description

REPLY: Thank you for suggestion. We agree that assignment of Miller indices needs more explanation. We have added additional information (in brackets) about Millers indices, assignment in our manuscript, in the part 2.2, Characterization methods:

For the structural analysis, the unit cell parameter a is related to the short macromolecule axis (correlated to planarity and side chains) and c corresponds to the long axis of polymer (length of the macromolecule), while b is related to the π-stacking period [25].

Following this logic, (00c) peaks, especially (001) peak, should be assigned to higher d-spacing values than (a00) peaks, while next (00c) peaks are 1/2 and 1/3 of (001) d-spacing, respectively.

  1. 4. Line 482: The author claimed: both carbazole-consisting imines (PAz-Carb) have shown only a peak corresponding to a parameter, connected with the short polymer axis, and thus with the planarity of macromolecules. Is this “short polymer axis” correlated with side-chains or the width of the chain? How does the short polymer axis correlate with the planarity of the macromolecules? Any references for it?

REPLY: We agree that following sentence is not clear. We have rephrased the explanation:

The structure order, visible in form of peaks in WAXD patterns, is related with macromolecules planarity and intermolecular interactions, but is not always related with π-stacking peak. In case of thin films examination, presence of π-stacking peak shows highly ordered structure. In ordered structures, with few peaks visible, more planar molecules might be characterized by (100) peak position – higher d-spacing values shows higher macromolecule planarity [25,26]. After the assignment of Miller indexes, it has been noticed, that both carbazole-consisting imines (PAz-Carb) has shown only a peak corresponding to a parameter, connected with the short polymer axis, and thus with the planarity of macromolecules.

  1. Line 488: The author claimed: Polyazomethine substituted solely with octyloxy groups (PAz-BOO-Oct) has revealed the peak, corresponding to planarity of the molecule, shifted towards higher angles (8.2° 2θ), suggesting wider an axis dimensions, which may be connected to the presence of n-octyloxy side chains. What is “an axis”? If the author means “a axis”, then it is confusing. An increase in 2theta value corresponds to a shorter correlation length. What does the author mean by “a wider a axis dimension”?

REPLY: Thank you for pointing that out. It is our mistake and clearly, some kind of typo. It should be:

Polyazomethine substituted solely with octyloxy groups (PAz-BOO-Oct) has revealed the peak, corresponding to  planarity of the molecule, shifted towards higher angles (8.2° 2θ), suggesting more narrow a axis dimensions, which may be connected to the presence of n-octyloxy side chains.

  1. It is not clear how the authors define the observation of the “planar” structure from WAXD. In conjugated polymer systems, one way to determine it is by observing the pi-pi stacking peak, which is typically at ~20 – 25 degrees and is missing from the data presented here.

REPLY: Thank you for this comment. We agree that this matter has not been appropriately explained. We hope, that the added passage can give the  proper explanation of that:

The structure order, visible in form of peaks in WAXD patterns, is related with macromolecules planarity and intermolecular interactions, but is not clearly related with π-stacking peak. In the case of thin films examination, presence of π-stacking peak shows highly ordered structure. In ordered structures, with few peaks visible, more planar molecules might be characterized by (100) position – higher d-spacing values shows higher macromolecule planarity [25,26].

  1. The optoelectronic properties were measured in the blend system with PCBM, but the morphology for the blend system was not investigated. There is a lack of systematic study on the morphology-performance relationship.

REPLY: Incorporation of additional data, regarding blend morphology could indeed provide additional information regarding morphology-performance relationship. Unfortunately, the efficiencies of studied systems have been very low, and thus, such correlation could be affected by a considerable error. The main emphasis has been therefore put on optical and electrochemical studies.

Additionally, by our mistake, in the previous version of manuscript there was a wrong title of this part. Should be: 3.6 Preliminary photovoltaic activity tests

(not 3.4. Morphology of thin films).

Reviewer 2 Report

The study has synthesized three novel conjugated polyazomethines using polycondensation of diamines consisting of diimine system, with either 2,5-bis(octyloxy)terephthalaldehyde or 9-(2-ethylhexyl)carbazole-3,6-dicarboxaldehyde. An attempt has also placed to partially replace bulky solubilizing substituents with the smaller side groups that has allowed the authors to investigate the effect of supramolecular organization. The properties of the resulting compounds were examined by NMR and FTIR spectroscopies, as well as characterized by the thermogravimetric analysis, differential scanning calorimetry, cyclic voltammetry, UV–Vis spectroscopy and X-ray diffraction. Among several other findings using other characterization methods applied, it was shown that the polymers display good thermal stability and high glass transition temperatures. As such, the results reported are well-trusted, which may be reproduceable to certain degree of accuracy. Since the basic discussions made are of publication quality, I suggest acceptance of the work for possible publication. A minor suggestion to the authors is that they must carefully read their paper to remove all kinds of errors associated with their tabulated data, as well as grammatical errors and typos.

Author Response

Reviewer #2:

The study has synthesized three novel conjugated polyazomethines using polycondensation of diamines consisting of diimine system, with either 2,5-bis(octyloxy)terephthalaldehyde or 9-(2-ethylhexyl)carbazole-3,6-dicarboxaldehyde. An attempt has also placed to partially replace bulky solubilizing substituents with the smaller side groups that has allowed the authors to investigate the effect of supramolecular organization. The properties of the resulting compounds were examined by NMR and FTIR spectroscopies, as well as characterized by the thermogravimetric analysis, differential scanning calorimetry, cyclic voltammetry, UV–Vis spectroscopy and X-ray diffraction. Among several other findings using other characterization methods applied, it was shown that the polymers display good thermal stability and high glass transition temperatures. As such, the results reported are well-trusted, which may be reproduceable to certain degree of accuracy. Since the basic discussions made are of publication quality, I suggest acceptance of the work for possible publication. A minor suggestion to the authors is that they must carefully read their paper to remove all kinds of errors associated with their tabulated data, as well as grammatical errors and typos.

REPLY: Thank you very much for this comment. The whole paper has been checked for errors, and as we hope all appropriate language corrections have been introduced.

Reviewer 3 Report

This manuscript demonstrated the strategy to introduce planar structure for production of polymers with high electrochemical and optical properties by simple alternations from bulky to smaller side groups. The authors presented a variety of experimental methods for verification of planner properties that novel imines obtained. Along with conventional methodologies to analyze characteristics of organic material, such as 1H NMR or FTIR spectra, thermal, optical, and electrochemical approaches were conducted. Moreover, x-ray diffraction was utilized for further studies of morphology. The properties of synthesized materials in solvent condition were compared with those in the form of thin film for application toward organic solar cell.

Nevertheless, the matter of credence should be concerned because data that support pivotal theme of the manuscript are unreliable. The idea of electron transition, which explains the effect of rigidity of side chains or planarity, was constructed based on energy gaps derived from electrochemical measurements. The onset potentials of either oxidation or reduction reaction should be determined in precise. For example, the x intercept of the tangent line at the point where the ordinate equals to the half of peak value could be decided as the onset potential. However, the suggested onset potentials in Table 4 seem to be established arbitrarily and therefore the numbers do not correspond with Figure 3. No matter how considerate the interpretation could be, reliability could only be obtained from objective analysis of experimental data.

Additionally, I would like to suggest some amendments to improve completion of the manuscript. For Table 5 and Figure 4, the measurement of absorbance spectra should be conducted several times (n > 3) for addition of error bar to verify reproducibility. The graphs of thermal examinations, such as TGA or DSC, were omitted on the manuscript but inclusion of those as supplementary data seems more desirable.

Author Response

Reviewer #3:

This manuscript demonstrated the strategy to introduce planar structure for production of polymers with high electrochemical and optical properties by simple alternations from bulky to smaller side groups. The authors presented a variety of experimental methods for verification of planner properties that novel imines obtained. Along with conventional methodologies to analyze characteristics of organic material, such as 1H NMR or FTIR spectra, thermal, optical, and electrochemical approaches were conducted. Moreover, x-ray diffraction was utilized for further studies of morphology. The properties of synthesized materials in solvent condition were compared with those in the form of thin film for application toward organic solar cell.

Nevertheless, the matter of credence should be concerned because data that support pivotal theme of the manuscript are unreliable. The idea of electron transition, which explains the effect of rigidity of side chains or planarity, was constructed based on energy gaps derived from electrochemical measurements. The onset potentials of either oxidation or reduction reaction should be determined in precise. For example, the x intercept of the tangent line at the point where the ordinate equals to the half of peak value could be decided as the onset potential. However, the suggested onset potentials in Table 4 seem to be established arbitrarily and therefore the numbers do not correspond with Figure 3. No matter how considerate the interpretation could be, reliability could only be obtained from objective analysis of experimental data.

REPLY: Thank you very much for this comment. Indeed, the onsets presented in Table 4 do not correspond to these, visible in Figure 3. This is a technical glitch, for which we are very sorry. Addition of a break in negative potentials x axis has caused an artificial shift of CV curves in the positive potential region. This break has been now removed, and the values of onsets correspond to these presented in the table. Determination of the onsets has been repeated, to double check the values. The Figure 3 has been replaced.

Additionally, I would like to suggest some amendments to improve completion of the manuscript. For Table 5 and Figure 4, the measurement of absorbance spectra should be conducted several times (n > 3) for addition of error bar to verify reproducibility. The graphs of thermal examinations, such as TGA or DSC, were omitted on the manuscript but inclusion of those as supplementary data seems more desirable.

REPLY: The measurements of solutions and thin films absorption has been conducted using a spectrophotometer with diffraction grating, which works using a spectrum sweep, measuring absorption at points 0.5 nm apart, with a very good reproducibility. This apparatus performs UV-Vis transmission measurements (T%) with the high precision ± 0,1%, what means the accuracy of absorbance (A) measurements  ± 0.001 a.u. within the whole spectral range (200-900) nm. Addition of error bars to each point could deteriorate presentation of the absorption results.

DSC, TGA and DTG curves of studied compounds have been presented in Supplementary Information (Figure S1 and S2), and appropriate new references [41] and [42] in the main text have been introduced.

Round 2

Reviewer 1 Report

  1. The author explained: “Such difference may be due to the large amount of long n-alkyl chains, together with a moderate oligomer chain length.” It is still not clear why the Tg show a >200 °C difference. Structurally, PAz-BOO-OOct is similar to PAz-Carb-OOct, and in terms of molecular weight, PAz-BOO-OOCt is even higher than PAz-BOO-OMe.
  2. For the third question, if the unit cell parameter a is the side-chain direction, how does it correlate with the planarity? P3HT shows clear high ordered (a00) peaks, but the backbone is not very planar. If the polymer backbone is rigid, backbone scattering (00c) can be seen, which comes from the scattering contrast (electron density difference) along the backbone. It is still not easy for planar donor-acceptor type conjugated polymers to see backbone scattering (00c) except for NDI polymers. However, in this case, if there is a backbone scattering signal in PAz-BOO-OOCt, shouldn’t it occur at a much higher 2theta because the distance between two moieties along the backbone is so close? If not, then where is the scattering contrast from? Thus, the attribution of Miller indices here is very confusing.
  3. For questions 4 to 6, again, the explanation of WAXD result is not convincing. The planarity of the macromolecule is not directly related to (a00) peak.
  4. Given the low efficiency, the novelty of this work is not convincing without further understanding of the morphology-performance relationship.

Reviewer 3 Report

Previously, the problem of credence and reliability of data had been presented as the values of onset potentials derived from electrochemical measurements seemed arbitrary. The authors had successfully replaced the disordered graph to straightened out the misunderstandings. Therefore, the values in Table 4 concur with those presented in Figure 3 in the revised manuscript. Along with the major issue, some amendments were made for improvement of the integrity. Additional data of thermal examinations were attached as supplementary data and the properties of the polymers are well demonstrated utilizing a variety of methodologies.